# Understanding the Impact of Assistive Technology on Users’ Lives in England: A Capability Approach

**DOI:** 10.3390/bioengineering12070750

**Published:** 2025-07-09

**Authors:** Rebecca Joskow, Dilisha Patel, Anna Landre, Kate Mattick, Catherine Holloway, Jamie Danemayer, Victoria Austin

**Affiliations:** Global Disability Innovation Hub, University College London, London WC1E 6BT, UK; rebecca.joskow.23@ucl.ac.uk (R.J.); dilisha.patel@ucl.ac.uk (D.P.); k.mattick@alumni.ucl.ac.uk (K.M.); c.holloway@ucl.ac.uk (C.H.);

**Keywords:** assistive technology, disability, capability approach, impact, user-centered

## Abstract

This study presents an analysis of England’s 2023 national assessment of assistive technology (AT) access and use, with a particular focus on the qualitative impact of AT as described by users. It aims to address limitations in conventional AT impact assessments, which often prioritize clinical outcomes or user satisfaction, by offering a deeper account of how impact is experienced in everyday life. Drawing on data from a nationally representative survey of 7000 disabled adults and children, as well as six focus group discussions and 28 semi-structured interviews with stakeholders across the WHO 5Ps framework (People, Providers, Personnel, Policy, and Products), the study applies Amartya Sen and Martha Nussbaum’s Capability Approach to explore these experiences. Using inductive thematic analysis, we identify three main domains of user-reported impact: Functions and Activities (e.g., mobility, communication, vision, leisure, daily routines, and cognitive support), Outcomes (e.g., autonomy, quality of life, safety, social participation, wellbeing, and work and learning), and Lived Experience (e.g., access barriers, essentiality, identity and emotional connection, peace of mind, and sense of control and confidence). These findings offer a more user-centered understanding of AT impact and can inform the development of future measurement tools, research design, and government-led interventions to improve AT provision.

## 1. Introduction

With over 2.5 billion Assistive Technology (AT) users, AT plays a critical and increasingly recognized role in enhancing independence, quality of life, and participation in society for people with disabilities [1]. As such, quality measurement tools and assessments of AT’s impact are essential for identifying how to improve users’ experiences with AT [1,2]. The global report on AT published by WHO and UNICEF in 2022 asserted that “…measuring outcomes and impact is necessary to understand the benefits of AT and create evidence-based policies to ensure universal access to it” [1]. However, based on a 2023 scoping review which examined what AT outcome measures are currently available and are used [2], there is “a lack of uniformity and concordance in the instruments” used to assess the impact of AT, with no standardized framework that evaluates the impact of AT in users’ lives across studies, or from both a device and service provision perspective [2]. Current AT impact evaluation methods remain limited, primarily assessing:(1)Functional efficacy and device satisfaction (e.g., reliability, performance, ease of use)(2)Clinical and observable outcomes (e.g., mobility improvements, physical health)(3)Device or user-group specific outcomes (e.g., wheelchairs and mobility impairments)

This research acknowledges the importance of objective measurements and outcomes, as well as functionality and health indicators [2]. Moreover, it recognizes the need to assess both functional efficacy and satisfaction to mitigate frequent assistive device abandonment [2]. However, to capture a broader understanding of impact, user experiences in accessing AT as well as the psycho-social value of AT in people’s lives must be considered. When framed by the Capability Approach, impact should be understood in terms of whether AT enables people with disabilities to “…achieve the things that they value,” and supports human flourishing [3,4,5]. These dimensions, including “…understanding how people assess, feel about, and make sense of” their experiences, can significantly influence AT adoption, use, and retention [6,7]. The International Classification of Functioning (ICF) is a WHO-developed framework used to describe domains of functioning [8]. “Participation” in public life is one such domain, and is seen as a “critical part of psychosocial well-being” [2,8]. Yet there are very few existing impact measurement tools that focus on this domain [2]. Moreover, while valuable in certain contexts, device or user-group specific measurement tools can make it difficult to substantiate the impact of AT on a broader scale [2]. While AT is defined by the WHO as encompassing products as well as systems and services [1], current research on impact measurement tools rarely reflects experiences of service provision [2], as exemplified in existing work on AT impact measurement tools [2]:

The Quebec User Evaluation of Satisfaction with AT (QUEST 2.0) is the current most used measurement tool for assessing AT’s impact [2]. It evaluates user satisfaction with a specific device by assessing characteristics such as size, weight, safety, and effectiveness, along with service-related aspects like delivery, repairs, and follow-up [9]. QUEST examines the broader system and is applicable to a broad range of assistive devices, but is limited to functional efficacy and satisfaction-focused measurement [9].

The Psychosocial Impact of Assistive Devices Scale (PIADS), “…a 26-item, self-report questionnaire designed to assess the effects of an assistive device on functional independence, well-being, and quality of life,” is the most used AT measurement tool for evaluating psychosocial impact [2,10]. PIADS measures impacts, adaptability, and self-esteem [6]. Notably, PIADS appears to have a “significant power to predict important AT outcomes” [6,10]; however, it relies on predefined categories [6], limiting the exploration of service provision experience and deeper, unanticipated impacts that users attribute to their AT.

The Client-Oriented Scale of Improvement (COSI) is a tool specifically designed for assistive listening devices [11]. Once the hearing aid is fitted, users assess the level of change using a five-point scale, ranging from “worse” to “much better,” [11]. COSI offers a unique user-led approach based on individual needs, but is limited by its device-specific focus and does not explore broader, quality of life impacts [11].

These limitations underscore the need for standardized, comprehensive AT impact measurement tools [2] that:(a)Are applicable across different disabilities and assistive devices;(b)Incorporate deeper user-reported experiences through qualitative methods;(c)Assess broader quality-of-life and psychosocial outcomes;(d)Captures both a device and service provision experience perspective.

From a constructivist paradigm perspective, the impact of AT is best understood through the perspectives of those who use it [3,7,12]. The failure to incorporate local, lived-experience knowledge has long been recognized as a key factor in the proliferation of high-cost, inappropriate technologies and ineffective programs [13]. In response, participatory approaches are increasingly recognized as important for measuring impact [14,15], particularly in capturing the subjective, cultural, and relational dimensions of well-being and quality of life that conventional tools often overlook [7,14,16]. However, in practice, existing AT impact assessments continue to focus primarily on functional efficacy and numerical satisfaction ratings, failing to capture the lived experiences and multidimensional benefits or challenges associated with AT use [2]. To address this gap, this research advocates for the inclusion of a holistic, user-centered framework that complements existing tools by prioritizing the language, perceptions, and meanings users attribute to AT, when assessing impact. This also includes the need to systematically capture users’ experiences of accessing assistive technologies, which remains a significant gap in existing outcome measures. Understanding the barriers and enablers to access is essential for informing service provision and policy recommendations [2]. By adopting a qualitative approach grounded in constructivist and Capability Approach principles, this study seeks to extend AT impact assessment beyond conventional functionality-based metrics to reflect the lived realities of users’ experiences.

The study was designed to address two core research questions:**(RQ1)** What is the impact of AT on the lives of AT users?**(RQ2)** How can this understanding inform future recommendations?

## 2. Materials and Methods

### 2.1. Background

This study builds upon findings of the integrated England Country Capacity Assessment, conducted by the Global Disability Innovation Hub in collaboration with the Cabinet Office’s Disability Unit in 2023 [17]. The assessment aimed to evaluate the need and access to assistive technology (AT) in England, assess the country’s capacity to meet these needs, and examine the impact of AT on the lives of disabled people [17]. To assess these factors, the study integrated three WHO AT measurement tools the rapid AT Assessment (rATA); the Assistive Technology Capacity Assessment (ATA-C); and the impact of AT (ATA-I) [18] (see Appendix A for more information). The rATA is a standardized population-based household-level survey measuring the need for, access to, satisfaction with and barriers associated with AT [19]. Data from rATA surveys help to inform policy and programs aimed at improving global access to essential AT [19]. The ATA-C is a system-level tool to evaluate a country’s existing capacity to regulate, finance, procure and provide AT [20]. Data from the ATA-C can help develop targeted strategies and investments to bridge the gap between the need for and the provision of AT [20]. Finally, the ATA-I tool, while still in development, aims to collect information about the impact of AT on a person directly [18]. For the purpose of this study, questions related to impact were integrated into the rATA and ATA-C as described above.

Key Informant Interviews (KIIs) and focus group discussions were conducted for the ATA-C across the WHO Global Cooperation on Assistive Technology (GATE) 5P framework, which evaluates People, Policy, Products, service Provision, and Personnel in AT systems [18]. Notably, the study found that Assistive Products (APs) significantly improved quality of life for users [17], with 83% of disabled individuals reporting that their APs were “very important” at all times [17].

Building on these results, the present study dives deeper into data from the England Country Capacity Assessment, focusing on the subjective impact of AT on user’s lives and user experience of accessing AT. The study adopts an inductive thematic analysis approach to systematically examine how users described the impact of AT across multiple data sources. Through a user-centered lens, the research seeks to deepen the understanding of AT’s impact, complement global measurement tools for assessing AT’s impact, and inform future policy and practice.

### 2.2. Theoretical Framework

This research is grounded in Amartya Sen’s ‘Capability Approach’ (CA), which reframes how we measure wellbeing and human development away from purely gross domestic product measures. CA focuses instead on “*…people’s capabilities to choose what they can do and be*” [5], and has been applied to disability and assistive technology (AT) contexts, notably in the paper ‘Assistive Technology (AT), for What?’ [3]. Sen’s Capability Approach distinguishes between functioning (actual achievements) and capabilities (the freedom to achieve) [5]. In the AT context, this means evaluating not only whether assistive devices enable specific tasks but also whether they expand users’ agency, self-determination, and social participation [3,5,21]. Likewise, Austin & Holloway posit that “*AT should be understood as a mechanism to achieve the things that AT users value*”, toward human flourishing [3,5,21,22]. While this study primarily utilizes an inductive analytical approach, it was influenced by this theoretical lens insofar as the thematic coding sought to explore both the functions AT allows as well as the things people sought to do.

This study also draws on Martha Nussbaum’s interpretation of the Capability Approach, which outlines core human capabilities as essential for a life of dignity and emphasized the need to ask, “*What is each person able to do and to be?*” [23]. Nussbaum, in contrast to Sen, argued that individuals have essential capabilities which apply to all beings, and designed what she termed ‘thick vague’ [23,24] descriptions of these capabilities (which include a right to life, bodily integrity, political participation, etc.) [23]. Descriptions were developed for the Essential Capabilities by Nussbaum but are intentionally ‘thick’ —they offer a clear and compelling vision of the core elements of human flourishing. However, they are also considered ‘vague’ because they are not prescriptive and allow for contextual variation and diversity in how these capabilities are realized. We find this a useful methodology for understanding the impact of AT; our analysis incorporates an effort to draw out ‘**thick, vague**’ [23,24] descriptions of AT impact with the intention to allow for a richer understanding of how AT interacts not only with function but also with identity, emotions, aspiration, and constraint, in a way which can be tested and built upon by other authors through other datasets.

### 2.3. Positionality

Our research team brings between 3 and 25 years of research experience. We are all from global north populations, namely, the United Kingdom and the United States, but have worked in diverse international contexts. Some members identify as disabled, while others do not. The study’s use of an inductive coding approach relies on researchers’ subjective interpretation. While we used reflexivity, iterative coding, and triangulation to support analytical rigour, we do not claim neutrality. Hence, it is relevant that our shared belief is that assistive technology (AT) users should have access to the products and support they require to live a free and just life, which informed how we engaged with the data. We offer this statement to transparently reflect on the assumptions and values that influenced our work.

### 2.4. Data Analysis

The analysis followed a “constructivist paradigm”, which emphasizes understanding a phenomenon from the perspective of those experiencing it [12]. A reflexive, inductive thematic analysis was conducted using Braun and Clarke’s approach [25,26], a process in which themes and patterns are defined from the data [12]. The analysis was conducted in two iterative stages: (1) inductive coding of free-text survey responses, followed by (2) inductive coding of focus groups and interviews. Given the larger volume of survey responses, the thematic framework was primarily derived from the survey data, while focus groups and interviews provided insights that refined and contextualized themes.

#### 2.4.1. Stage 1: Thematic Coding of Survey Data

In the first stage, we conducted inductive coding of survey responses to identify common patterns and themes. The survey dataset included 6403 open-text responses to the survey question: “*Please describe how impactful your assistive products are to your life in general.*”

Using MAXQDA software v2020 (MAXQDA, VERBI Software, we generated preliminary summary transcripts using the MAXQDA AI Assistant which were carefully reviewed, cross-checked against original transcripts, and validated by the research team. We then systematically reviewed the responses and assigned descriptive codes to meaningful text segments. This initial phase allowed us to identify recurring words, phrases, and ideas, which were then grouped into broader thematic categories. Through the initial analysis, three overarching categories were identified (see Table 1): Functions/Activities, Outcomes, and Lived Experience Impact. Functions/Activities and Outcomes addressed *what* the impact of AT was, as described by users. Functions/Activities focuses on the specific functions and activities that AT enables, such as mobility, communication, or daily living tasks. Outcomes reflect the broader benefits or goals facilitated by AT, such as independence, employment, or social participation. Lived Experience Impact examined *how* users related to and described the impact of their AT, including the meanings they assigned to it, the language they used, and their reflections on their experience accessing and using it.

Under **Functions/Activities**, responses related to AT enabling activities of daily living (ADLs) were coded under the subtheme “*Daily Living/Routine*”, including tags such as “*getting dressed*,” “*errands*,” and “*hygiene*.” Under Outcomes, responses related to users expressing their AT enabling autonomy, including tags such as “*sense of choice*”, “*reduced reliance on caregivers*”, *or* “*freedom*”, were grouped under the subtheme “*Autonomy/Independence*”. Under Lived Experience Impact, excerpts related to the “*enormity*” of the impact of AT on their lives, their “*reliance*” or “*dependence*” on their AT, or the “*frequency of use*” of AT was categorized under the subtheme “*Essentiality/Dependency*”. For example, if a user described a mobility aid as essential to their daily life because it gives them independence to move around, these aspects were coded as follows:**Functions/Activities** →“*Mobility*”**Outcomes** → “*Independence*”**Lived Experience** → “*Essential*”

As part of an iterative refinement process, MAXQDA tools were used to examine code frequency and distribution, helping to identify the most prevalent themes. Related categories were consolidated to streamline the coding framework. For example, “*communication*” was grouped with “*hearing*” under “*Communication/Hearing*”, and “*vision*” was merged with “*reading*” under “*Vision/Reading*”. Similarly, themes related to personal significance, identity, and emotional attachment were consolidated under “*Identity and Emotional Connection*”.

Some descriptive codes, initially nested within broader subthemes, were reclassified due to their prominence. For example, while “*safety*” was initially coded solely as an Outcome (reflecting practical benefits such as reducing fall risks), it also appeared in emotional contexts (e.g., peace of mind, vulnerability without AT). As a result, “*Peace of Mind*” was added as a distinct subtheme under Lived Experience Impact.

Additionally, MAXQDA was leveraged to extract user testimonials and highlight significant patterns, further supporting the validation of emerging themes.

#### 2.4.2. Stage 2: Thematic Coding of Focus Groups and Interviews

In stage two, we continued to refine and validate the coding framework using six focus group discussions and 28 interviews. Many themes aligned across the datasets, and the coding framework was strongly validated. Focus group discussions and interviews also provided further and more detailed narratives, which were extracted to contextualize the subthemes around areas that were difficult to capture in brief survey responses, such as “*Absence Framing*”, “*Work/Learning*”, and “*Access Barriers/Unmet Needs*”. Throughout this process, we took steps such as triangulating across all data sources, keeping an audit trail with notes on coding, theme development, and framework refinements, and employing reflexivity throughout the coding and iteration processes, to ensure methodological rigor and validity, and minimize research biases [26].

#### 2.4.3. Ethical Considerations

Favorable ethical approval was obtained from University College London [Ref: 24371.001]. All participants were aware of the research objectives before participating and were able to freely withdraw from the research. We provided multi-modal consent procedures to suit accessibility needs. The survey followed WHO ethical approval procedures, where only anonymized data was provided for analysis. Ethical standards were rigorously maintained throughout the research.

## 3. Results

This section presents key subthemes and findings from the study, as they pertain to how users perceived and described the impact of AT on their lives. Thematic analysis identified three primary themes, **Functions/Activities, Outcomes**, and **Lived Experience Impact** each encompassing six subthemes, summarized in Table 1.

### 3.1. Functions/Activities

This section summarizes the primary activities and functions that participants reported as directly enabled by AT, which were mobility, communication and hearing, vision and reading, leisure and hobbies, daily living and routine, and cognitive support.

#### 3.1.1. Mobility

Participants frequently highlighted the critical role of AT in facilitating mobility. Many described how their mobility aids, such as wheelchairs, crutches, walking sticks, and pressure relief cushions, were essential for balance, wayfinding, leaving the house, and navigating their environment:

“*My disability makes life difficult mostly due to my mobility problems. Without my assistive products, my life would be much more difficult and severely restricted. […]. Having a scooter and my crutches is essential. There are so many things I simply couldn’t do without them.*”(Survey)

“*I’m unable to walk without my stick and couldn’t go out without my wheelchair so my mobility aids are life changing.*”(Survey)

#### 3.1.2. Communication and Hearing

AT was commonly reported as a key facilitator of communication, including self-expression, hearing, and understanding. Users highlighted how devices such as communication boards, hearing aids, and text-to-speech applications allowed them to participate in social interactions:

“*Using my communication board, I’ve been able to express myself for the first time in years.*”(Participant 10, Interview)

“*The ability to hear again with my aids made me feel like I was a part of the world again.*”(AT ‘People’, Focus Group Discussion)

#### 3.1.3. Vision and Reading

For many participants, AT such as screen magnifiers, glasses, and adaptive reading tools played an essential role in enabling them to engage with the world visually, whether for reading, work, or daily tasks and pursuits:

“*I use magnifiers every day to read books, recipes, and even recognize faces.*”(Participant 11, Interview)

“*Glasses allow me to read, see the world, and be independent.*”(Survey)

“*I could not see properly without glasses, drive, read, watch tv, use computer etc. Everything really.*”(Survey)

#### 3.1.4. Daily Living and Routine

AT was often cited as essential for managing day-to-day tasks and used throughout users’ everyday routines, assisting with managing pain and medication, personal care and hygiene, as well as running errands. It also assisted with the operation of household devices such as TVs, intercoms, door openers, and lighting systems:

“*I rely on many assistive products every day—to get out, to see, to wash, to remind me to take medication. I’d be lost without them.*”(AT ‘Provision’, Focus Group Discussion)

“*I couldn’t carry out my daily life without [AT].*”(Survey)

“*[AT] relieve[s] my back pain and allow me to complete daily tasks.*”(Survey)

“*You know, I read the Quran in English, in Arabic with an app on my phone, and it’s because of that that I can actually take part in something that’s really important to me*”(AT ‘Product’, Focus Group Discussion)

“*The continence products keep me dry, and help prevent accidents*”(Survey)

#### 3.1.5. Leisure and Hobbies

Many participants emphasized AT’s role in facilitating enjoyment of leisure activities, hobbies, and relaxation, such as such as gardening, knitting, gaming, and creative arts:

“*Assistive devices let me enjoy activities I thought I’d never do again, like gardening.*”(Participant 24, Interview)

“*I can enjoy hobbies like knitting with my adapted tools.*”(Survey)

“*[AT] allow[s] me to engage in recreational activities, like playing games with friends.*”(Survey)

“*I use my assistive device for painting, which brings me joy.*”(Survey)

“*I can now sing in my choir and hear music and television, doorbell etc, and help me living on my own*”(Survey)

#### 3.1.6. Cognitive Support

Many participants highlighted the importance of AT, such as digital task managers and reminder systems, in supporting cognitive function, particularly in organizing daily life, staying focused, and managing memory-related tasks:

“*I use reminders and organizers to manage my day effectively.*”(Survey)

“*My cognitive aids help me remember to take my medication on time.*”(Survey)

“*[AT] helps me stay focused and oriented, and to not lose track of where I am going.*”(Survey)

### 3.2. Outcomes

This theme captures the broader personal and social benefits facilitated by AT, as reported by users. The identified subthemes were autonomy and independence, quality of life, comfort and safety, social participation and inclusion, health and well-being, and work and learning.

#### 3.2.1. Autonomy and Independence

Many participants emphasized the role of AT in fostering independence, especially in terms of freedom and choice, reducing their reliance on caregivers, and providing a sense of control over their environments:

“*[AT] Allow(s) me some independence and freedom to help look after myself and feel more human*”(Survey)

“*Without these [assistive products] I would not be able to live independently or do any of the things I do*”(Survey)

“*The ability to use assistive devices to control my environment, from opening doors to managing lights and appliances, makes me feel more independent and capable of living alone*”(AT ‘Product’, Focus Group Discussion)

“*[AT] help[s] me to retain my dignity and independence to a large degree.*”(Survey)

#### 3.2.2. Quality of Life

AT was widely recognized as positive and improving overall quality of life. Users describe AT as a lifeline, reducing stress, expanding opportunities, and increasing participation in society:

“*Assistive technology has completely transformed my quality of life, allowing me to do things I never thought possible. From social participation to being able to manage my health better, it’s a lifeline*”(AT ‘Product’, Focus Group Discussion)

“*Without a wheelchair I am completely housebound. I held off asking for one for a long time as I was embarrassed, but it’s meant I can get out and do/see things and take part in the world now, as I couldn’t before. […] I can’t empty my bladder without catheters, so I would be very, very ill without them!*”(Survey)

“*Makes life a pleasure*”(Survey)

#### 3.2.3. Comfort and Safety

Participants highlighted AT’s role in reducing discomfort, preventing injuries, and ensuring safety. Many described their reliance on AT for fall prevention, comfort, and increased security, both at home and in public spaces.

“*Keep me safe whilst walking in the house. Wheelchair keeps me safe when outside*”(Survey)

“*My assistive products changed my life completely. I feel much safer going out and being in the house in general*”(Survey)

“*Without [AT] I wouldn’t be mobile [and] would not be safe showering*”(Survey)

“*The assistive products that I use allow me to reduce the friction on my feet, which suffer from palmoplantar pustulosis. Without the main product that I use, […] I’d have much more friction from walking and thus more pain. They allow me to walk further, stand for longer and basically do more than I would without them.*”(Survey)

#### 3.2.4. Social Participation and Inclusion

AT was described as an essential tool for social interaction, allowing users to engage with family, friends, and their broader communities. Participants emphasized the importance of AT in reducing isolation and fostering inclusion in work and social settings:

“*[AT] allows me to go to work, see my friends and family and communicate when I’m struggling*”(Survey)

“*[AT] means that I can meet and speak with people*”(Survey)

“*[AT] makes me feel included in my social and work life circle.*”(Survey)

#### 3.2.5. Health and Wellbeing

AT was frequently attributed to improved physical and mental health. It was described as relieving pain, improving mood, and enhancing overall wellbeing:

“*[AT has] improved my mental and physical health.*”(Survey)

“*[AT] alleviate[s] my pain and enhance[s] my overall well-being.*”(Survey)

“*[Assistive] Products help with tasks that provide comfort and improve mood.*”(Survey)

#### 3.2.6. Work and Learning

Participants reported that AT was an essential tool for reducing barriers employment and education, and in turn, helping them to achieve their goals:

“*I wouldn’t be able to do my job without [AT].*”(Survey)

“*Very important, I can’t see without [AT]. [AT] helps me learn like everyone else*”(Survey)

### 3.3. Lived Experience Impact

This section explores how users frame and describe the role and significance of AT in their lives, including the language they use and the meanings they attribute to AT, as well as their reflections on their experiences accessing and using AT. The identified subthemes were access barriers and unmet needs, absence framing, essentiality and dependence, identity and emotional connection, peace of mind, and sense of confidence and control.

#### 3.3.1. Access Barriers and Unmet Needs

Where users expressed frustration or dissatisfaction, it was not necessarily directed at the assistive product itself. While some concerns related to device repair or a lack of personalized or adaptable design, frustrations were more often focused on the lack of or fragmented support (e.g., training, lack of repair), accessibility (e.g., discrepancies in access, funding), or better systems surrounding AT (e.g., long wait times):

“*Assistive Technology has changed my life… but the system lets it down. I feel like you have to fight so hard to get choice, and sometimes it’s just not worth the effort*”(Participant 25, Interview)

“*It’s not the kit that’s the problem. It’s the process to get into the system in the first place… We didn’t even know who to call, and when we did, they said we’d have to wait eight weeks. That was for something as simple as a pendant alarm*”(Participant 4, Interview)

“*… there’s a lot of new technology out there, but often it would be helpful to have training. Training in how to use it and what’s available. Because it can be a bit overwhelming, but it is essential obviously because as [a] disabled person, […] if you’re not really techy it can be a bit of a barrier to get used to lots of different new devices and computer software and what not*”(AT ‘People’, Focus Group Discussion)

“*For me a lot of the things more to do with systems than with the technologies themselves necessarily. So there used to be centres for independent living where you could go and try a whole lot of different things and sometimes you really do have to try things to see if they are going to work*”(AT ‘People’, Focus Group Discussion)

Even with AT, some users felt their needs were only partially addressed:

“*I’ve struggled to get the device repaired, which leaves me feeling abandoned.*”(AT ‘Provision’, Focus Group)

“*I need more help to use it effectively.*”(Survey)

“*It works, but it’s not designed for someone like me*”(Survey)

Users shared their perception that better systems could enhance the impact of AT:

“*The waiting lists are ridiculous—by the time you get what you need, you’ve already suffered for months*”(Participant 19, Interview)

“*I think what might help in Social Services is listening more to people with lived experience and to have co production. So, it’s like an equal power around policies and decisions and service delivery. I think it would improve the system so that if disabled people and blind people were involved in the very beginning of the design of the systems*”(AT ‘People’, Focus Group Discussion)

#### 3.3.2. Essentiality and Dependence

Many users describe AT as “essential” and “needed,” describing AT as a non-negotiable fundamental requirement, rather than a luxury or convenience:

“*These devices are critical; they are not luxuries—they’re the difference between living and merely existing*”(Participant 18, Interview)

“*I am totally reliant on [my hearing aids]*”(Survey)

“*They are essential for every aspect of my life […]*”(Survey)

In fact, users frequently framed the importance of AT by imagining life without it, describing it as indispensable to their daily lives. Many stated that without AT, they would be unable to leave their homes, work, or engage in everyday life activities.

“*If I receive no help from assistive products, it means I don’t go out, I don’t read, write, draw. I cannot hear well, and I am lost in a world of my own*”(AT ‘Provision’, Focus Group)

“*Without [my assistive technology,] I wouldn’t be able to see properly, walk without pain in my lower limbs or remember to take my medication*”(Survey)

“*Without my hearing aids, I would be isolated*”(Survey)

“*Life would be impossible without [AT]*”(Survey)

“*Couldn’t live and thrive without [AT]*”(Survey)

“*Life would be unbearable without [AT]*”(Survey)

“*Without [AT], I would be just like a prisoner in my own home*”(Survey)

This highlights users’ reliance on AT, as without it, they would face a profound loss of independence and engagement with the world around them.

#### 3.3.3. Identity and Emotional Connection

Users often emphasized the emotional, social, and psychological significance of AT. Many described their devices as extensions of their body or identity, deeply integrated into how they understood themselves and their roles in the world.

“*A wheelchair provision may mean getting back to work and family roles, reducing pain, and maintaining physical symmetry. These devices impact anatomy, physiology, and emotional roles like identity and purpose*”(Participant 22, Interview)

“*My assistive products are like a part of my body*”(Survey)

“*My grabber is my third arm*”(Survey)

“*My assistive products feel like a part of my body—they’re integral to who I am and how I live…*”(Participant 14, Interview)

Recognizing that AT transcends is not just a tool, but an integral part of users’ identity and way of being, is essential to fully understanding its impact.

#### 3.3.4. Peace of Mind

Users described AT as offering emotional relief, reducing anxiety, and providing a sense of reassurance and security:

“*My devices make me feel secure. Without them, I’d be lost and anxious about simple daily tasks*”(Participant 10, Interview)

“*I feel safe knowing my assistive devices are reliable. They help me go about my day without constantly worrying about what could go wrong*”(AT ‘Provision’, Focus Group Discussion)

“*I would feel vulnerable and in constant danger without them*”(Survey)

“*…the emotional relief that these products provide—knowing I can rely on them—is immeasurable*”(Participant 14, Interview)

AT can serve as a conduit for reducing anxiety and cultivating a sense of security and reliability in users’ daily lives.

#### 3.3.5. Sense of Control and Confidence

Users describe how AT enables them to feel more self-reliant and a sense of control, allowing them to go about their daily routines and participate in social interactions and public life with greater self-confidence:

“*The fact that I can control my lights, communicate with my caregiver, and watch TV using one device has transformed my daily life. It’s not just functional—it’s emotional, making me feel less reliant and more capable*”(Participant 24, Interview)

“*Assistive devices […] also give me the confidence to interact socially and reduce my anxiety about being out of the house.*”(Participant 24, Interview)

“*Gives me a little bit of courage to leave the house*”(Survey)

“*Very impactful—they give me the confidence to know that when I am in flare or having a migraine I could still function to some degree*”(Survey)

AT can empower users with greater self-reliance and sense of control, fostering the confidence needed to fully participate in public life and social interactions.

## 4. Summary: ‘Thick, Vague’ Descriptions of AT Impact

Thematic analysis revealed that users describe the impact of assistive technology (RQ1) across three key domains: the specific functions and activities it enables; the broader life outcomes it supports; and the lived experiences through which users reflect upon the impact of AT. Below, we summarize the evidence to offer the ‘thick vague’ description of each domain:

### 4.1. AT Enabling Functions and Activities (What AT Allows Users to Do)

The “Functions/Activities” category refers to the tangible functions and activities that assistive technology (AT) directly enables. These include both basic functional actions—such as moving, seeing, hearing, or communicating—and activities that go beyond function, such as pursuing hobbies, preparing meals, or staying in touch with friends and family. These are the things that may be directly enabled by AT itself. For instance, using a cognitive support device to stay focused (a function/activity) may allow a person to participate in school or manage household responsibilities (outcomes). Similarly, using AT to bathe or use the toilet (a function/activity) may enable autonomy and safety (outcomes). Using a mobility aid (such as a cane, walker, or wheelchair) to leave the house (a function/activity) may lead to greater health/wellbeing and social or community participation (outcomes). The distinction between what AT enables directly and what it makes possible more broadly or over time is important to understanding its full impact.

### 4.2. AT Enabling Outcomes (What Broader Life Goals AT Makes Possible)

This category refers to what those functions and activities make possible in terms of broader life outcomes. These are the indirect impacts, often life goals or fulfilment of needs, that result from being able to engage in those functions and activities. These are not things AT does directly—but rather what doing those things enables for the person. Outcomes may reflect what becomes possible (or impossible) in someone’s life when they are able (or unable) to perform certain activities: autonomy and independence; physical health; emotional wellbeing; peace of mind; confidence; social participation and inclusion; dignity, purpose, and identity; the ability to contribute, work, or learn; and improved quality of life. As one user shared: “I use my assistive device for painting, which brings me joy.” Here, *painting* is the activity. *Joy* is the outcome (e.g., which is categorized under the “Health and Wellbeing” outcome in the subthemes in this analysis). It is what the function or activity makes possible in the person’s life.

### 4.3. Lived Experience of AT (How the Impact of AT Is Experienced and Felt)

This category goes beyond understanding what AT enables in user’s lives, both directly and indirectly, to capture how users experience and interpret these impacts, including the language they use to describe it, the relationships they form with their devices, and the emotional and existential significance they attach to the presence or absence of AT in their lives. Users frequently described AT as essential and life-changing, with one user even attributing AT to “*the difference between living and merely existing*”. One common way users conveyed impact was through absence framing—imagining or remembering life without AT to highlight its significance, reflections which often emphasized restriction, invisibility, or isolation. In contrast, users describe how the presence of AT offered peace of mind, reduced feelings of vulnerability, and enabled a greater sense of control and confidence to go out into the world and pursue their goals. Many users described a personal or emotional connection to their devices, influencing identity and sense of self. Some users even described their device as being “*like a part of my body*.” At the same time, users spoke of the toll of the systems aspect of AT, describing fragmented services, access barriers, limited choice, and bureaucratic delays that made it difficult to obtain or maintain the technology. These challenges were often accompanied by feelings of frustration, exhaustion, and inequality, which in some cases diminished the overall benefit of AT.

These thick vague descriptions are reflective of this data, and it will be interesting to understand if they hold for other datasets in other contexts, given that its sample was of disabled people only, and in the high-income setting of England.

## 5. Discussion

This discussion interprets the findings regarding the impact of assistive technology (AT) on users’ lives within the context of the literature. Through a ‘thick vague’ impact lens, we reflect on the future implications of the findings (RQ2). Limitations and potential directions for future research are also outlined.

### 5.1. Implications

Currently published AT impact guides and tools only capture certain elements of AT’s impact—such as service delivery and system-level factors (e.g., QUEST [9]), psychosocial aspects (e.g., PIADS [10]), or user satisfaction with the device itself (e.g., COSI [11]) [2], as such, they often fail to capture the full picture of AT’s impact. As argued by White and Pettit, “generating numerical values to represent [user’s] assessments […] does not necessarily reflect the way that people live their lives, or capture the underlying rhythms within which they take action and understand the meaning of their experience overall” [7]. These tools may address fragments of the “what”, such as functions, activities, and certain outcomes, but they often miss the outcomes that are truly important to users, including broader personal, emotional, and social benefits. Critically, they rarely address the “how”—how AT is experienced within broader systems of support, how it shapes a person’s sense of self and identity, how it provides a sense of control and peace of mind, how it enables people to go out confidently, and how it allows people to imagine and pursue future possibilities. Yet, these lived experiences are central to understanding AT’s true impact. Specifically, they fall short in illustrating the full picture of how AT can enable human flourishing, as framed by the *AT for What* framework, which proposes: “Human flourishing as an operational framework for disability justice:“ [3] with AT positioned “as a (vital and important) mechanism to achieve broader aims (outcomes), linking to people’s capabilities and freedoms to choose what they can do and be” [3]. Building on this, our paper contributes to expanding both the discourse and the practical tools needed to realize the goal of human flourishing through assistive technology—a goal grounded in the *AT for What* framework—and offers a foundation for others to build upon.

Moreover, our user narratives suggest that AT impact is shaped as much by context and support systems the device and user are embedded in as by the device itself. These systems—encompassing access, training, repair, affordability, and policy—can either enable or constrain a person’s ability to flourish. Measuring the often-burdensome impact of the AT system on the individual living within it, and the design of the system itself as a potential barrier to accessing devices, is important; documenting systemic failures and user frustrations can spotlight areas where investment in access, training, repair, or affordability is urgently needed. Without capturing these elements, measurement tools risk underestimating AT’s value and undermining how robust, person-centered evidence can inform responsive policies and justify funding. Thus, we argue that the lived experiences and self-defined goals of AT users should play a role in shaping how impact is measured, understood, and acted upon.

This study highlights the kinds of value that traditional indicators often miss yet are critical to users’ lives. Such an approach can complement global measurement tools, like WHO ATA-I, by demonstrating how participatory, user-driven evidence can provide a deeper and more accurate understanding of AT’s real-world impact, revealing what matters most to users and highlighting outcomes not captured by standard metrics. It can also generate rich user narratives that can shape policies and communication tools that speak to the priorities and realities of users, raise awareness through compelling storytelling, as well as inform and strengthen the case for investment in AT with concrete examples and case studies. In general, participatory methods can also be seen as “vehicles for […] policy influencing, engagement, and advocacy” [7] as well as for “bridging gaps between diverse actors at different levels” and creating a sense of ownership [27] (e.g., involving stakeholders across the 5Ps such as with the England Country Capacity Assessment [17]).

We acknowledge that in lower-resourced settings, service providers are often focused on immediate delivery needs, and thus, impact assessment can sometimes be perceived as a “luxury problem”. However, we posit that understanding AT impact from a user perspective remains a critical piece of the puzzle—not only to ensure that scarce resources are used effectively and equitably, but also to shape interventions around outcomes that matter most to users in their specific contexts.

### 5.2. Limitations

While the rATA is an internationally validated tool on the basis of its content and construct validity [28], it has some limitations. As a self-reported measure, it captures individuals’ experiences and perceptions of the impact of assistive products, but these responses may be influenced by participants’ awareness of assistive products, their uses, benefits, or limitations [17]. The rATA functional assessment is based on the Washington Group Short Set, which does not capture psychosocial disability [29]. In effect, disability and AT needs may be underestimated if intersectional and diverse groups are not well represented [29]. Moreover, while participatory methods offer many benefits as described, it should be cautioned that they can be exploited and “[…] used to obscure differences within target communities, legitimize extractive and exploitative processes of information gathering, impose external agendas, and contain or co-opt potential popular resistance” [7].

Moreover, subjective, narrative-based approaches—while critical for centering lived experience—should not replace standardized metrics but rather should be seen as a complementary layer of insight. Flexibility and longitudinality are needed to adapt to changing user needs and contexts, and to track impact over time.

Furthermore, this analysis is based on data from England, a high-income country. Therefore, findings may not be generalizable to regions with different cultural, economic, or healthcare frameworks. For instance, in some contexts, greater stigma surrounding assistive technology and disability may shape perceptions differently and create barriers. Greater stigma may limit a person recognizing their own needs, seeking services, or adopting assistive technology [30]. Environmental factors can also present barriers to the adoption of AT, whether through prevailing negative attitudes or inaccessible physical environments. The use and impact of AT must therefore be understood within the environment and context of a specific user. Micro-level studies cannot simply be “scaled up” to inform macro-level insights without careful consideration of local context [7]. For instance, while this study focused on the England context, it is important to acknowledge that assistive technology must be appropriate not only to the delivery system, but also to the broader environmental context in which it is used. Although environmental fit was not a prominent theme in our findings, it remains a critical factor, particularly in settings with complex terrain or inaccessible infrastructure.

### 5.3. Future Research

We hope that future research will build on and refine this approach to understand the impact of AT in other settings. Moreover, while this analysis primarily focused on the human impact of AT e.g., “AT as a vital mechanism to achieve the things users value” [3], societal and economic impact are also important elements of impact measurement [31], and necessitate further exploration. Future research could explore economic and societal benefits, including: Employment rates among AT users versus non-users [31]; Reduction in healthcare costs due to prevention of secondary health issues [2]; Increased engagement with public infrastructure by AT users [2]; Cost savings from enhanced independence (e.g., reduced reliance on subsidies, support programs, or caregivers) [31]. Expanding impact assessment frameworks to include economic indicators will provide stronger evidence to inform funding, policy decisions, and global AT investment strategies. As one user noted, “*If we calculate the social return on investment, providing assistive technology means someone with disabilities can lead a better life and contribute to society*” (Participant 9, Interview).

## 6. Conclusions

These findings underscore that a user-centered, mixed-methods approach, one that includes both functional and psychosocial outcomes such as quality of life, wellbeing, and participation, is essential to fully understand the impact of assistive technology (AT). Such an approach is essential to understanding how AT contributes to human flourishing, as framed by the *AT for What* framework. The WHO Assistive Technology Assessment (ATA) toolkit, developed through the Global Cooperation on Assistive Technology (GATE) initiative, offers a valuable mechanism to capture users’ experiences, including the barriers and supports that shape access. These factors are critical to understanding the broader impact of AT. The ATA toolkit also presents an opportunity to standardize how we assess AT impact in a way that reflects both lived experience and social outcomes. While there is currently no agreed framework or shared language for assessing AT impact, this paper offers an early attempt to group and describe it across three key domains within one national context. Advancing the understanding and measurement of AT impact will require collaboration across sectors and disciplines, including with AT users, policymakers, and researchers. We offer this contribution as a foundation for that ongoing conversation, and as a step toward more inclusive, responsive, and justice-oriented AT systems.

## Figures and Tables

**Table 1 bioengineering-12-00750-t001:** Key Themes of Assistive Technology (AT) Impact (Note: This table presents key themes of assistive technology (AT) impact across three broad dimensions. These categories are not intended to represent direct pairings or exclusive groupings. Many themes interact across multiple dimensions.).

Question	Impact Dimension	Themes
***What* is the impact of AT, according to users?**	**FUNCTIONS/ACTIVITIES** What AT directly enables users to do in their daily lives	Mobility Communication and hearing Vision and reading Leisure and hobbies Daily living and routine Cognitive support
	**OUTCOMES** The broader benefits or goals supported by AT	Autonomy and independence Quality of life Comfort and safety Social participation and inclusion Health and wellbeing Work and learning
***How* do users experience the impact of AT?**	**LIVED EXPERIENCE** How users describe, relate to, and make sense of AT’s impact in their lives	Access barriers and unmet needs Essentiality and dependence Identity and emotional connection Peace of mind Sense of control and confidence

## Data Availability

Requests for anonymized data should be made to the corresponding author.

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
