# Peer review of "Understanding the Impact of Assistive Technology on Users’ Lives in England: A Capability Approach"

_bioengineering, 2025, doi:10.3390/bioengineering12070750_

Round 1
Reviewer 1 Report
Comments and Suggestions for Authors
It's a pleasure reading this well written and relevant paper. The content is interesting and relevant. However from the perspective of many regions and users in less privileged countries many issues may seem like ‘luxury problems’ in an ‘utopian vision’ of AT provision. Such a perspective should be debated in the discussion. Developing AT impact assessment is clearly important, but only when a certain context has put in place and internalized the need for such assessment. The endless waiting lists of many public health care systems is a symptom of resources being focused on ‘putting out fire’ wherefore AT impact assessment may be extremely relevant for theoretically optimizing resources, but leaving a practical gap.
The language is fine but please consider to make parts like abstract and conclusion more inclusive (a clear thought can be expressed clearly with simple words according to Wittgenstein)
In the following are some specific comments
Background
As the tools are relatively new and incompletely published please provide an overview of the content of rATA, ATA‑C and ATA‑I
Please expand abbreviations like “England CCA”
I do not understand the purpose of this phrase here ln - 616‘We welcome further development of this thinking from other researchers.’
Please review psychometric validation metrics for your statement ‘While the rATA is an internationally validated tool [15],...’ to avoid misinterpretations.
Consider revisiting the context around ‘surrounding assistive technology and disability may shape perceptions differently.’ which is very important and therefore merits a slightly clearer logical flow and can bear a longer discussion. Especially what follows seems ‘deus ex macchina’
Speaking of which, you may want to consider a more inclusive language throughout your article. Expressions like ‘writ large’ are beautiful like latin or chinese characters, but are they inclusive for a wider audience? That brings me to the final point - please consider an appendix reporting a synthesis of the WHO toolkit used (rATA,ATA-I, etc)
Discussion & Conclusion are both numbered 5
Author Response
Thank you very much for taking the time to review this manuscript. Please find the detailed responses below and the corresponding revisions/corrections highlighted in red in the re-submitted files
Comment 1:
However from the perspective of many regions and users in less privileged countries many issues may seem like ‘luxury problems’ in an ‘utopian vision’ of AT provision. Such a perspective should be debated in the discussion. Developing AT impact assessment is clearly important, but only when a certain context has put in place and internalized the need for such assessment. The endless waiting lists of many public health care systems is a symptom of resources being focused on ‘putting out fire’ wherefore AT impact assessment may be extremely relevant for theoretically optimizing resources, but leaving a practical gap.
Response 1:
Thank you for your valuable reflection. We recognize the concern that, from the perspective of many lower-resourced regions, discussions around AT impact assessment may appear to reflect a utopian vision—especially when overwhelmed with immediate delivery demands and service gaps. We have added text to the discussion at line 625 to acknowledge this tension. However, we also posit that understanding AT impact from a user perspective is a necessity, even in resource-constrained settings, though it represents one piece of a larger puzzle. Centering the lived experiences and priorities of users can help ensure that even limited resources are directed toward interventions that meet the most critical needs. Impact assessment can also support more equitable and strategic allocation of resources and contribute to long-term system strengthening. We have revised the discussion to reflect this position and better acknowledge the practical realities raised in your comment.
Comment 2: The language is fine but please consider to make parts like abstract and conclusion more inclusive (a clear thought can be expressed clearly with simple words according to Wittgenstein)
Response 2: Thank you for this helpful suggestion. We agree with the comment regarding language clarity and inclusiveness. Accordingly, we have revised the abstract (line 9) and conclusion (line 649) to use simpler, more direct language. This includes reducing jargon, shortening complex sentences, and more clearly articulating concepts to ensure accessibility to a wider audience.
Comment 3: Background As the tools are relatively new and incompletely published please provide an overview of the content of rATA, ATA‑C and ATA
Response 3: Thank you for pointing this out. A brief overview of what these tools comprise of and how they were used for the purpose of this study have been included in the 2.1 Background at line 125.
Comment 4: Please expand abbreviations like “England CCA”
Response 4: Thank you for pointing this out. We agree with this comment. We have now expanded the abbreviation “England CCA” to “England Country Capacity Assessment” in all relevant locations (lines 135 and 616). Similarly, we have spelled out “World Health Organization’s Global Cooperation on Assistive Technology (GATE)” at first mention (lines 130 and 693). Acronyms are now only used where they are clearly introduced in the same paragraph.
Comment 5: I do not understand the purpose of this phrase here ln - 616‘We welcome further development of this thinking from other researchers.’
Response 5: Thank you for this helpful comment. The original sentence was intended to signal that our categorization of AT impact is an early, exploratory effort, not a definitive framework, and that we encourage further development and testing by other researchers as well as in different contexts. To make this clearer, we have removed the sentence from its original location and repositioned it within the future research section (line 637). We have also revised the phrasing for clarity.
Comment 6: Please review psychometric validation metrics for your statement ‘While the rATA is an internationally validated tool [15],...’ to avoid misinterpretations.
Response 6: Thank you for flagging this potential area for misinterpretation. We have now specified the rATA has been internationally validated with respect to content and construct validity, and added the appropriate citation which further describes that this was done by multiple international working groups of experts.
Comment 7 : Consider revisiting the context around ‘surrounding assistive technology and disability may shape perceptions differently.’ which is very important and therefore merits a slightly clearer logical flow and can bear a longer discussion. Especially what follows seems ‘deus ex macchina’
Response 7: Thank you for your reflections on this point. We agree this is an important aspect to develop and have therefore added some greater context from line 663. This additional content, we hope, reflects how AT use needs to be understood within the context – and how stigma and environmental barriers will influence the impact of AT.
Comment 8 : Speaking of which, you may want to consider a more inclusive language throughout your article. Expressions like ‘writ large’ are beautiful like latin or chinese characters, but are they inclusive for a wider audience?
Response 8: Thank you for this helpful suggestion. We agree that some expressions, such as “writ large”, may not be inclusive for all readers. To address this, we have paraphrased the sentence to preserve its meaning while using more accessible language. The revised sentence can be found at line 632.
Comment 9: That brings me to the final point - please consider an appendix reporting a synthesis of the WHO toolkit used (rATA,ATA-I, etc)
Response 9: Thank you for your comment. We have added a synthesis of the WHO toolkit as well as a link to more information in Appendix A.
Comment 10: Discussion & Conclusion are both numbered 5
Response 10: Thank you for flagging this. I changed Conclusion (649) to be numbered 6.
Reviewer 2 Report
Comments and Suggestions for Authors
This article develops a deeper, more nuanced look at evaluating the impact of AT. As AT is vital for the limiting exclusion among persons with disabilities, this is an important topic. The paper is very well done and presented. I only have a few comments.
1) I don't really see the need for creating the "absence" framework. It is just the negative of other aspects of the framework. Granted it is a different way of expressing these things, but I don't really see it adding anything to the analysis.
2) Maybe the paper could be a little clearer at making the distinction between AT not being designed or delivered well, versus limitations that might be a result of the environmental context. This might not be as big of a deal in a high income country, but in a low income country it could be that a high quality wheelchair, well-delivered and maintained may be of limited benefit because the terrain is such that the wheelchair doesn't make as big an impact on their life. It could be that a cochlear implant has less of an impact in an area where hearing loops are not available. Obviously these things impact the effectiveness of AT and should be incorporated in the analysis but they are somewhat distinct from the AT delivery system. It would be good to clearly delineate where in the ecosystem problems may arise
3. Table 1 appears to say that only communication and hearing impact quality of life and only mobility affects autonomy, etc. This can't be right, so there is something about the presentation of this table that needs to be changed.
Author Response
Thank you very much for taking the time to review this manuscript. Please find the detailed responses below and the corresponding revisions/corrections highlighted in red in the re-submitted files.
Comment 1: I don't really see the need for creating the "absence" framework. It is just the negative of other aspects of the framework. Granted it is a different way of expressing these things, but I don't really see it adding anything to the analysis.
Response 1: Thank you for your thoughtful comment. We included this theme as it came up quite frequently from participants describing “without AT x, y, z” when asked to describe impact (e.g., prompting participants to imagine life without AT). We do see that it reiterates the broader theme of "dependency and essentiality" and have now removed it from the broader framework, instead incorporating the findings under that theme at line 455.
Comment 2: Maybe the paper could be a little clearer at making the distinction between AT not being designed or delivered well, versus limitations that might be a result of the environmental context. This might not be as big of a deal in a high income country, but in a low income country it could be that a high quality wheelchair, well-delivered and maintained may be of limited benefit because the terrain is such that the wheelchair doesn't make as big an impact on their life. It could be that a cochlear implant has less of an impact in an area where hearing loops are not available. Obviously these things impact the effectiveness of AT and should be incorporated in the analysis but they are somewhat distinct from the AT delivery system. It would be good to clearly delineate where in the ecosystem problems may arise
Response 2: Thank you for this helpful comment. We agree that assistive technology (AT) must be considered in relation not only to its delivery system but also to the broader environmental context in which it is used. While this paper focuses on the England context—where such environmental considerations were less prominent in our data—we acknowledge that environmental fit remains a critical factor for AT effectiveness, particularly in other settings. To reflect this, we have added a few sentences to the limitations section in line 632.
Comment 3: Table 1 appears to say that only communication and hearing impact quality of life and only mobility affects autonomy, etc. This can't be right, so there is something about the presentation of this table that needs to be changed.
Response 3: Thank you for this helpful comment. We agree that the previous layout of Table 1 may have unintentionally suggested one-to-one relationships between themes. To address this, we have combined tables 1 and 2 to improve clarity, and restructured the table by organizing the themes vertically under each impact dimension (Functions/Activities, Outcomes, and Lived Experience). We have also added a clarifying footnote to explicitly state that themes are not intended as fixed or exclusive groupings. The revised version of Table 1 can be found on line 262 of the manuscript.